# Innate Immune Response to Powassan Virus Infection: Progress Toward Infection Control

**DOI:** 10.3390/vaccines13070754

**Published:** 2025-07-15

**Authors:** Mohammad Enamul Hoque Kayesh, Michinori Kohara, Kyoko Tsukiyama-Kohara

**Affiliations:** 1Department of Microbiology and Public Health, Faculty of Animal Science and Veterinary Medicine, Patuakhali Science and Technology University, Barishal 8210, Bangladesh; 2Transboundary Animal Diseases Center, Joint Faculty of Veterinary Medicine, Kagoshima University, Kagoshima 890-0065, Japan; 3Department of Microbiology and Cell Biology, Tokyo Metropolitan Institute of Medical Science, Tokyo 156-8506, Japan; kohara-mc@igakuken.or.jp

**Keywords:** Powassan virus, toll-like receptor, immune response, vaccine, toll-like receptor agonist, adjuvants

## Abstract

Powassan virus is an emerging tick-borne flavivirus that poses a significant threat to human health. The outcome of Powassan virus infection is shaped by both viral factors and the host immune response. While this review aimed to examine the innate immune response, particularly toll-like receptor-mediated immune responses to Powassan virus, data specific to the immune response to Powassan virus remain scarce. Therefore, we focused on toll-like receptor responses to related flaviviruses to infer possible mechanisms of host response. Insights from both in vivo and in vitro studies are critical for guiding the development of effective therapeutic and preventive strategies. Currently, there are no clinically approved treatments or vaccines for Powassan virus, highlighting the urgent need for their development. We also highlight recent progress in POWV vaccine development, with an emphasis on the potential use of toll-like receptor agonists as adjuvants to enhance immunogenicity and improve vaccine efficacy.

## 1. Introduction

Powassan virus (POWV), a reemerging neurotropic tick-borne flavivirus of increasing public health concern, belongs to the genus *Flavivirus* in the family *Flaviviridae* [1]. POWV was first isolated in 1958 from the brain of a five-year-old child who died from fatal encephalitis in Powassan, a small town in northern Ontario, Canada [2]. The virus is primarily transmitted by *Ixodes scapularis*, *Ixodes cookei*, and other *Ixodes* tick species to small- and medium-sized mammals. Humans serve as accidental dead-end hosts due to inadequate viremia to infect new ticks [3]. In a mouse model, POWV was successfully transmitted within 15 min of *Ixodes scapularis* attachment [4].

It persists in nature through a cycle involving infected ixodid ticks and mammals such as mice, woodchucks, and skunks [5]. A notable case of blood transfusion-transmitted POWV infection was reported in a kidney transplant recipient who likely contracted the virus through blood transfusion [6]. Experimental milk-borne transmission of POWV has been reported in goats [7]. POWV strains have been isolated from different animals, including squirrels, woodchucks, dogs, foxes, horses, spotted skunks, white-footed mice, birds, and mosquitoes, and antibodies have been detected in other animals, including monkeys, hamsters, chickens, and rabbits [8]. Wildlife can act as reservoirs for POWV and pose a threat to humans and domestic animals [9].

POWV is a positive-sense, single-stranded RNA virus with a genome of approximately 11 kb, encoding a single long open reading frame, which is processed into three structural proteins, the capsid (C), premembrane (prM), and envelope (E) proteins, and seven nonstructural proteins, NS1, NS2A, NS2B, NS3, NS4A, NS4B, and NS5 [10]. The virus can cause various clinical presentations ranging from asymptomatic or mild cases to severe neuroinvasive diseases [11]. The incubation period for POWV infection also varies and is generally 1–5 weeks [12].

POWV has two distinct genetic lineages: lineage I, the prototype POWV, and lineage II, known as the deer tick virus (DTV) [13]. POWV and DTV share 84% nucleotide and 94% amino acid sequence identity [14]. Based on the envelope protein sequence, POWV and DTV share 96% amino acid sequence identity and are identical in terms of serology and clinical presentation [15,16]. It is believed that the DTV diverged from the POWV approximately 200 years ago [15]. Differences in clinical presentation, pathology of the central nervous system (CNS), and immune responses between lineages in immunocompetent mouse models have been reported [17]. Both lineages can cause CNS infections in humans. Infection from tick bites can result in fatal encephalitis with mortality rates of 19.1% in adults and 7.1% in children. More than 50% of survivors experience long-term neurological sequelae [18], which has also been observed in a mouse model [19]. Other neurotropic flaviviruses that can cause encephalitis, cognitive impairment, seizure disorders, and paralysis include the West Nile virus (WNV), Japanese encephalitis virus (JEV), tick-borne encephalitis virus (TBEV), and Zika virus (ZIKV) [20].

Sporadic human infections with POWV have been reported in the United States, Canada, and Russia [13,21]. Although the incidence of human POWV infection remains low, the number of detected cases has increased over the past decade. This increase may be associated with several factors, including enhanced surveillance systems, improved laboratory diagnostic facilities, expansion of the vector population, and increased exposure of humans [13,21,22,23]. As the climate continues to warm, POWV vectors have become widely distributed [24,25,26], further contributing to the expansion of their populations. Notably, because of its often-fatal outcomes, a better understanding of the virus is imperative for developing preventive and therapeutic strategies.

POWV is a growing public health concern, particularly in North America, and no preventive vaccines or effective treatments are currently available for POWV infection. Therefore, a safe and effective vaccine is urgently required. Although POWV infections are sporadic and localized, climate change-driven shifts in tick populations may increase the number of human cases. Combined with ecological changes and potential viral mutations, POWV could emerge as a significant global health threat [27]. Compared to other pathogenic flaviviruses that infect humans, POWV remains one of the less studied human pathogenic flaviviruses [28]. Understanding the detailed mechanisms of host-virus interactions is critical for developing therapeutic and preventive interventions.

The innate immune system is an important component of host immunity, playing a key role in sensing invading pathogens, including viruses, and subsequently activating adaptive immunity, resulting in the early control of viral infection [29,30,31,32]. Pattern recognition receptors (PRRs), which are encoded in the germline, play a key role in detecting conserved microbial components known as microbe-associated molecular patterns (MAMPs) and pathogen-associated molecular patterns (PAMPs). These include several receptor families such as Toll-like receptors (TLRs), RIG-I-like receptors, NOD-like receptors, C-type lectin receptors, AIM2-like receptors, and various DNA-sensing receptors [33,34].

Among the various PRRs, TLRs are the most extensively characterized. TLRs play a central role in the early detection of pathogens by recognizing PAMPs, leading to the induction of proinflammatory cytokines and chemokines. This response helps in shaping the outcome of infection by priming the immune response and bridging innate and adaptive immunity [35,36]. However, the specific role of TLRs in POWV infection remains poorly understood. Although TLRs are essential for initiating host defense, their activation can act as a double-edged sword—dysregulated or excessive TLR signaling may contribute to immune-mediated pathology rather than protection [37]. Modulating TLR responses may therefore offer a promising strategy for developing novel therapeutic and preventive interventions against POWV infection. Understanding how TLRs interact with vaccine antigens could inform the design of vaccines that elicit more robust and long-lasting immunity. Consequently, a detailed understanding of TLR involvement in POWV infection is critical for both immunopathogenesis research and the development of therapeutic or preventive interventions. In this review, we discuss the TLR-mediated immune response to POWV infection, recent progress in POWV vaccine development, and the potential use of TLR agonists as vaccine adjuvants to enhance the efficacy of vaccines.

## 2. TLR Response to POWV Infection and Mechanism of Immune Evasion

Toll-like receptors (TLRs) are implicated in the early interaction between host cells and invading viruses, regulating viral replication and/or host responses, and ultimately modulating viral pathogenesis [38]. Upon recognition of viral proteins or RNA, the corresponding TLRs become activated and recruit specific adaptor proteins, initiating downstream signaling cascades that lead to the production of type I interferons and proinflammatory cytokines (Figure 1). Acronyms used in the Figure 1 are defined in the figure legend. MyD88 is the primary adaptor protein for most TLRs, except for TLR3, which signals exclusively through TRIF. TLR4 is unique in that it utilizes both MyD88 and TRIF to mediate downstream signaling [39].

TLRs activate multiple steps in inflammatory reactions that help eliminate invading pathogens and coordinate systemic defenses. TLRs play an important role in shaping adaptive immunity [40]. TLRs commonly implicated in the immune response against viral infections include TLR2, TLR3, TLR4, TLR6, TLR7, and TLR8 [41]. TLR9 is involved in detecting DNA containing unmethylated CpG motifs [42]. TLR10 has been reported to participate in the detection of viral proteins from the influenza virus [43]. Activation of TLRs against arboviral infections induces an immune response, leading to increased viral clearance and protection against disease occurrence. In contrast, the inhibition of TLRs can reduce arbovirus infection-associated inflammation and tissue damage. The modulation of TLRs represents a potential therapeutic strategy to combat arbovirus infections [41].

TRAF6 (TNF receptor-associated factor 6), an E3 ubiquitin ligase, plays a critical role in the NF-κB (nuclear factor kappa-light-chain-enhancer of activated B cells) signaling pathway and in various PRRs-mediated signaling cascades [44]. In the TLR signaling pathway, TRAF6 leads to the activation of interferon regulatory factors (IRFs), leading to the production of interferons and contributing to the antiviral response (Figure 1). Despite its antiviral functions, TRAF can also exert proviral effects. For example, interaction between the NS3 protease of tick-borne flavivirus and TRAF6 has been shown to enhance replication of the tick-borne flavivirus [45]. However, whether a similar interaction occurs between TRAF6 and NS3 protein of POWV, and whether it influences POWV replication, remains to be determined.

Important roles of TLRs against flavivirus infections, including dengue virus (DENV) [46], WNV [47], and ZIKV [48], have been documented. Activation of TLR signaling is essential for mounting an effective antiviral response. However, overactivation or dysregulation of TLR signaling can lead to excessive inflammation, resulting in tissue damage, cytokine storms, poor disease outcomes, and increased viral persistence and immune evasion [49,50]. Differential TLR responses have been reported in flavivirus infections [46,51]. TLR3 plays a dual role in flavivirus infections. For example, TLR3 expression exerts an antiviral effect in DENV and WNV infections, whereas TLR3 expression enhances ZIKV replication [46,47,51]. However, TLR7 plays a protective role against DENV and other flavivirus infections [46,52].

It is unclear which PRRs are dominant in POWV, and the role of TLRs in POWV infection remains poorly understood, requiring further investigation. POWV infection induces both innate and adaptive immune responses in the host. As an RNA virus, POWV can activate different PRRs, including TLRs (e.g., TLR3, TLR7, and TLR8) and RIG-I-like receptors (RLRs; e.g., melanoma differentiation-associated gene 5 [MDA5] and RIG-I), resulting in the production of cytokines and interferons (IFNs) [53].

Cytosolic PRRs, RIG-I and MDA5 play key roles in the detection of a broad range of RNA viruses, including flaviviruses [54]. Upon recognition of viral RNA, RIG-I and MDA5 become activated and interact with mitochondrial antiviral signaling protein (MAVS), triggering downstream signaling pathways that culminate in the production of type I interferons [54,55]. Flaviviruses, such as DENV and ZIKV, utilize the NS4A protein to disrupt the interaction between RIG-I and MAVS, thereby suppressing antiviral signaling [56,57,58]. Various non-structural proteins of flaviviruses antagonize the interferon signaling pathway by promoting the degradation of STING or inhibiting the activation of TBK1 and IKKε, thereby facilitating viral infection [59,60]. The NS5 protein of several flaviviruses has been identified as a potent inhibitor of IFN signaling [61]. Studies in RIG-I-/- and MDA5-/- mice have demonstrated higher infection susceptibility with RNA viruses compared to control mice, suggesting the antiviral functions of these sensors against RNA viruses, including flaviviruses [62]. However, the association of RIG-I and MDA5 in the recognition of POWV and their roles in modulating POWV infection remain to be elucidated.

A recent RNA-Seq study showed that both POWV and DTV elicit strong interferon responses; however, POWV-infected tissues exhibited significantly higher expression of NLRP6 and IL1α compared to DTV-infected tissues [17]. In contrast, DTV-infected tissues show significantly elevated expression of CXCL3, CD177, and CAV1 (caveolin-1), indicating enhanced neutrophil recruitment and T cell–dependent NF-κB activation [17]. During flavivirus infection, TRIM5α inhibits viral replication by binding NS2B/3 protease, promoting its K48-linked ubiquitination and subsequent proteasomal degradation [35]. An increased level of TRIM5 was also observed in infection with POWV and DTV [17].

Manipulation of mitochondrial processes to evade the innate immune response has been studied in many flaviviruses, including ZIKV and DENV; however, it has yet to be examined in the case of POWV infection [63]. Tick saliva can exacerbate viral infections by modulating host immune responses and promoting viral entry and spread [64]. In a previous study, saliva-activated transmission with a low dose of POWV was reported. However, the specific salivary factors responsible for enhancing POWV transmission are yet to be elucidated [65].

The skin remains the first line of defense and is key to host–pathogen–vector interactions [66]. Infiltration of neutrophils and mononuclear cells is observed in POWV-infected ticks compared to uninfected ticks, and macrophages and fibroblasts have been identified as early cellular targets of infection at the tick feeding site [28]. During viral inoculation, modulation of proinflammatory cytokines and chemokines at the skin interface has been reported [67]. At 3 h post-inoculation (hpi), inflammatory cytokines and chemokines, including interleukin (IL)-1B, IL-6, TLR-4, and chemokine receptor type 3 (CCR3) are upregulated, resulting in an increased number of phagocytes and neutrophils [67].

POWV can cause structural aberrations in the dendrites of neurons, which was also observed in the brain tissue of POWV-infected mice [68]. POWV induces fewer cytokine responses and less detectable apoptosis than WNV infection [68]. In another study in a murine model, comparative transcriptomics of brains infected with POWV lineage I or lineage II demonstrated the activation of different immune pathways and downstream host responses, suggesting differences in host immune responses and disease pathogenesis between lineages [17]. A recent study reported immunomodulation during POWV transmission at the skin interface in *Ixodes scapularis* ticks, characterized by neutrophil recruitment and interleukin signaling [69].

Mlera et al. reported that the intraperitoneal and intracranial inoculation of 10^3^ PFU of POWV in a 4-week-old white-footed mouse (*Peromyscus leucopus*), a rodent native to North America, caused no overt clinical signs of disease and showed an early immune response [5]. In contrast, intracranial inoculation of 4-week-old C57BL/6 and BALB/c mice was lethal at 5 dpi, whereas intraperitoneal inoculation was lethal in BALB/c mice, but 40% (2/5) of C57BL/6 mice survived, suggesting the involvement of restriction factors [5]. However, the mechanism by which *P. leucopus* mice limit POWV infection remains unclear [5]. Overall, data on the innate immune response to POWV infection are limited, highlighting the need for further investigation.

## 3. POWV Vaccine Development and the Role of TLR Agonists as Adjuvants

Currently, treatment of POWV infection is supportive, and no FDA-approved vaccines or specific therapeutics are available [70]. Due to this unmet medical need, efforts are underway to develop different types of POWV vaccine candidates for clinical use, including live-attenuated [71], nucleic acid [72], viral-vectored, and subunit vaccines [73,74]. The structural proteins of POWV—capsid, premembrane (prM), membrane (M), and envelope (E)—are being explored as key antigens for vaccine design [75]. Human sera from individuals who received the tick-borne encephalitis virus (TBEV) vaccine, experienced natural infection, or were infected despite prior vaccination, have shown cross- protection against several members of the tick-borne encephalitis complex, such as Kyasanur Forest disease virus and Alkhumra virus. However, cross-neutralization of POWV following TBEV vaccination is minimal [76]. A recent bioinformatics study identified and characterized B-cell and T-cell epitopes from POWV proteins, leading to the design of a stable and immunogenic vaccine candidate [75]. Computational analyses predicted strong immune interactions, supporting its potential efficacy and highlighting the need for further experimental validation [75]. Cheung et al. recently developed a live-attenuated chimeric vaccine candidate using a YFV-17D vector expressing POWV prM/M and E proteins. By removing the NS1 glycosylation site in the YFV-17D nonstructural protein, they further attenuated the vector to improve safety. In mice, a two-dose regimen conferred 70% protection, while a heterologous prime-boost strategy using the EDIII protein achieved 100% protection, underscoring the potential of this approach for effective POWV vaccination [71].

Modern adjuvants, such as TLR agonists, are engineered to activate specific TLRs, mimicking PAMPs to enhance vaccine efficacy. Through TLR engagement, these agonists promote dendritic cell maturation, improve antigen presentation, induce cytokine and chemokine production, and modulate the T helper (Th) cell responses [77]. Due to their potent immunostimulatory properties, TLR agonists have emerged as promising adjuvants in vaccine development, offering both immunomodulatory and immunotherapeutic benefits [78,79,80,81]. Their ability to tailor immune responses has shown promise in enhancing the efficacy of viral vaccines, including those targeting flaviviruses such as POWV [48]. Selecting an appropriate TLR agonist depends on multiple factors, including disease pathogenesis, the desired immune response, host species and age, route of administration, and antigen characteristics [82]. TLR agonists are especially valuable for vaccine platforms that require additional immune activation, such as virus-like particle (VLP) and nucleic acid-based vaccines, positioning them as attractive candidates for next-generation vaccines, including those targeting flaviviruses like POWV. The potential applications and considerations for TLR agonists in vaccine development are summarized in Table 1.

VLPs resemble viruses but lack the genetic material required for viral replication, making them a safe platform for vaccine development [83]. VLPs mimic the viruses from which they are derived in terms of size, geometry, and ability to activate T-helper cells [84,85]. VLP-based vaccines are safer alternatives to inactivated or attenuated viruses, which may pose health risks if not fully inactivated or attenuated. VLP-based vaccines have been approved for use against various viral infections, including hepatitis B virus (HBV) [86], human papillomaviruses (HPV) [87], and hepatitis E (HEV) [88]. VLPs induce strong humoral responses and cellular immunity, including Th1 and cytotoxic T lymphocyte (CTL) activation, making them ideal antigens for future tick-borne flavivirus vaccines [89,90]. VLP-based vaccination has also been proven to be a safe and effective platform for developing a POWV vaccine, as demonstrated by previous studies [70,91].

VLP-based vaccination strategies have demonstrated safety and efficacy in the context of POWV. A candidate vaccine consisting of prM and E proteins induced complete seroconversion and high levels of neutralizing antibodies in immunized mice, providing protection against POWV infection [70]. This approach also protected mice against lethal POWV challenge, with robust B and T cell responses contributing to immune protection [91]. Identifying additional immune-protective factors will aid in the rational design of next-generation POWV vaccines.

Recent studies have also highlighted the potential of TLR agonists to enhance the efficacy of VLP-based vaccines [92]. Crawford et al. reported that POW-VLP vaccines adjuvanted with novel synthetic TLR 7/8 agonist INI-4001 or TLR4 agonist INI-2002 induced significantly higher levels of POWV-binding and neutralizing antibodies compared to VLP alone or with the conventional alum adjuvant [92]. Notably, INI-4001 outperformed INI-2002, enhancing antibody response, reducing POWV neuroinvasion, and providing complete protection against a lethal challenge with POWV in mice vaccinated with low doses of POWV-VLP [92]. Despite these promising findings, the precise immunological mechanisms underlying the enhanced protection remain to be fully elucidated [92].

Malonis et al. developed a lumazine synthase–based nanoparticle immunogen displaying domain III (EDIII) of the POWV E glycoprotein [73]. Lumazine synthase has been reported to function as a TLR4 agonist, activating TLR4 signaling pathways that regulate both innate and adaptive immune responses, including dendritic cell maturation and CD8^+^ T cell cytotoxicity [93]. In mice, the EDIII-displaying nanoparticle elicited strong EDIII-specific neutralizing antibodies and conferred protection against POWV infection, providing valuable insights into antibody-mediated protection, and guiding future EDIII-based vaccine strategies [73].

mRNA vaccines hold significant promise for interrupting zoonotic disease transmission through cross-species immunity, which may protect both human and animal health by reducing tick populations and the spread of tick-borne pathogens, fostering a healthier global ecosystem [94]. A recent study has demonstrated that ionizable lipid nanoparticles (LNPs), used to deliver mRNA vaccines, activate innate immune pathways through TLR4 signaling, even in the absence of mRNA [95]. This TLR4-mediated activation can enhance vaccine efficacy by stimulating key transcription factors such as NF-κB and IRF, highlighting the adjuvant potential of LNPs in mRNA vaccine platforms. Van Blargan et al. developed a LNP-encapsulated modified mRNA vaccine encoding the POWV prM and E genes, which proved highly immunogenic in mice, inducing potent neutralizing antibodies and providing protection against lethal challenge from both POWV lineages [96]. Furthermore, the vaccine elicited a cross-reactive antibody response against Langat virus, another tick-borne flavivirus [96], suggesting broader neutralizing potential.

A synthetic DNA vaccine encoding the POWV prM and E genes elicited a robust humoral response that was specific to POWV antigens and conferred protection against lethal challenge in mice [72]. This vaccine also produced POWV-specific T cell response, contributing cellular immunity against envelope antigens [72]. Incorporating TLR agonists as adjuvants could further enhance the immunogenicity of this DNA vaccine by promoting stronger innate and adaptive immune activation [97]. To facilitate the development and evaluation of novel therapeutics and preventive strategies, it is essential to develop suitable animal models that recapitulate the natural course of POWV infection and clinical manifestations observed in humans.

## 4. Discussion

TLRs and RLRs are key components of the innate immune system, responsible for sensing viral infections and initiating antiviral responses [98]. Their involvement in both host defense and the pathogenesis of flavivirus infections is increasingly evident [99]. However, their specific role in POWV infection remains poorly defined, highlighting the need for further research. Differences in TLR expression across cell types and species further complicate our understanding of host–virus interactions, as these variations can influence susceptibility and disease outcomes [100]. Notably, TLR signaling can exert both protective and pathogenic effects: while appropriate activation supports antiviral defense, dysregulation or overactivation may exacerbate inflammation and tissue damage [37,101].

Additionally, individual TLRs may elicit divergent responses to the same viral family. For instance, TLR3 has been shown to both enhance and limit ZIKV infection, underscoring its context-dependent role [52]. The dual role of TLR3 underscores its complexity as a therapeutic target, requiring precise modulation to enhance protection while avoiding harmful effects [102]. This duality makes it essential to characterize TLR responses during POWV infection to determine their contributions to viral control or disease progression. Understanding how POWV modulates or evades TLR-mediated signaling could also reveal novel mechanisms of immune evasion and identify potential therapeutic targets. For instance, many flaviviruses disrupt interferon pathways downstream of TLR activation, a strategy that may also contribute to POWV’s neuroinvasive capacity and persistence in the CNS. Future studies should focus on the interactions between POWV and key RNA-sensing TLRs—particularly TLR3, TLR7, and TLR8—to better define their roles in host defense and viral pathogenesis.

TLR agonists have been successfully used as vaccine adjuvants to enhance immunogenicity, especially in the context of viral infections [48]. Their use as experimental vaccines against related flaviviruses has shown promise in inducing robust and balanced immune responses [48]. Accordingly, incorporation of TLR agonists into POWV vaccine formulations may enhance their effectiveness. However, such an approach requires careful selection of the TLR agonist, validation of its safety and efficacy in preclinical models, and a final evaluation in clinical settings. Additionally, growing evidence suggests that the LNPs and mRNA components in LNP-encapsulated mRNA vaccines may act synergistically to activate TLR pathways, thereby enhancing innate immune responses [103]. However, while such immune stimulation may improve vaccine effectiveness, it also carries the risk of increased inflammation and potential adverse effects, as LNPs alone have been shown to induce proinflammatory responses in animal models. The combination of multiple TLR agonists may amplify innate and adaptive immune activation through synergistic or additive effects, potentially allowing for lower adjuvant doses with improved safety and cost-effectiveness [104,105]. Nonetheless, combinatorial adjuvant strategies present several challenges, including the risk of excessive or uncontrolled inflammation, induction of immune tolerance, and complexities in optimizing dosage and delivery mechanisms. Moreover, the outcome of combining TLR ligands is highly context-dependent; it remains unclear whether ligands that engage the same intracellular signaling pathways produce synergistic or antagonistic effects. Therefore, a more comprehensive understanding of the cellular targets and immune pathways activated by different TLR agonists is crucial for the rational design of adjuvant combinations that can safely elicit desired immune responses, ultimately facilitating the development of more effective and precisely tailored vaccines.

## 5. Conclusions

A balanced TLR response is essential for achieving protective immunity while minimizing immunopathology. Understanding the interplay between TLR signaling pathways is particularly important in the context of POWV infection, where the host immune response remains poorly characterized. Currently, all POWV vaccine candidates are in preclinical stages, highlighting the need for accelerated clinical development. TLR agonists hold promise for enhancing the immunogenicity of POWV VLP-based and DNA vaccines; however, their safety, efficacy, and mechanisms of action require further investigation.

## Figures and Tables

**Figure 1 vaccines-13-00754-f001:**
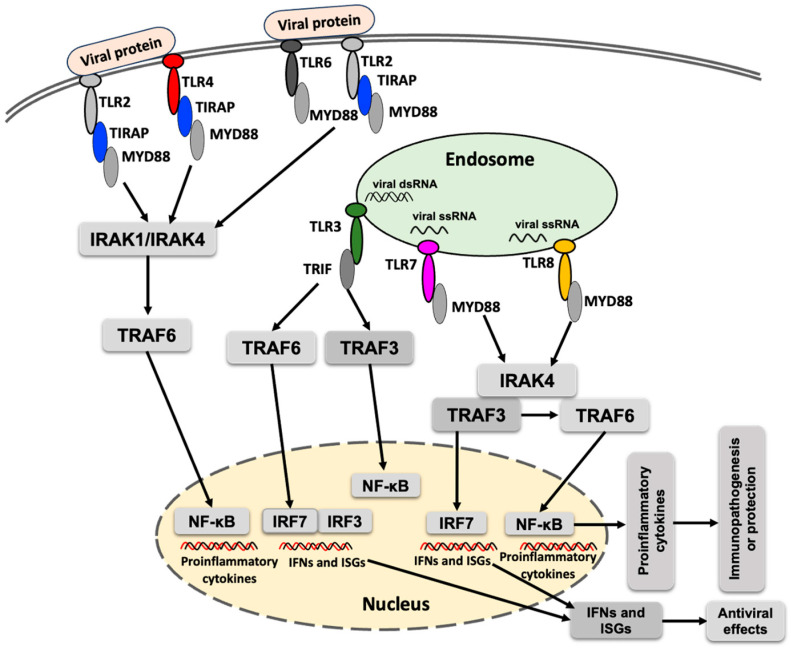
A simplified schematic presentation of TLR response to flavivirus infection. TLR signaling is activated upon detection of the viral proteins or nucleic acids, which culminates in the induction of proinflammatory cytokines, interferon-stimulated genes (ISGs) and interferons. MyD88 (myeloid differentiation primary response 88), TIRAP (toll-interleukin 1 receptor domain–containing adaptor protein), TRIF (TIR domain–containing adaptor–inducing IFN-β), IRAK1 (interleukin-1 receptor-associated kinase 1), IRAK4 (interleukin-1 receptor-associated kinase 4), TRAF3 (TNF receptor–associated factor 3), TRAF6 (TNF receptor-associated factor 6), NF-κB (nuclear factor kappa-light-chain-enhancer of activated B cells), IRF3 (interferon regulatory factor 3), IRF7 (interferon regulatory factor 7), IFN (interferon), POWV (Powassan virus).

**Table 1 vaccines-13-00754-t001:** Overview of the role and potential of TLR agonists in vaccine development, with implications for POWV vaccines.

Aspect	Description
Role of TLR agonists	Trigger TLRs to activate the innate immune response, promoting immunity against infections.
Function as vaccine adjuvants	TLR agonists function as potent immunological adjuvants, enhancing the efficacy and immunogenicity of vaccines.
Vaccine research	Provide immunomodulatory and immunotherapeutic effects; serve as a new toolbox in vaccine research.
Pathogen-specific optimization	Tailored TLR agonists have shown enhanced protection against various pathogens, including flaviviruses.
POWV vaccine development	Limited data available; however, emerging data suggest that incorporation of TLR agonists in POWV vaccine formulations can significantly improve immunogenicity and efficacy, particularly for VLP-based and DNA vaccine candidates.
Future directions	Integration of TLR agonists with multi-epitope-based vaccines, designed using bioinformatics tools, may be a promising strategy to elicit potent neutralizing antibodies.

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
