# Peer review of "Innate Immune Response to Powassan Virus Infection: Progress Toward Infection Control"

_vaccines, 2025, doi:10.3390/vaccines13070754_

Round 1
Reviewer 1 Report
Comments and Suggestions for Authors
In the manuscript, Kayesh et al. aim to review the innate immune response to Powassan virus (POWV) infection. POWV is a reemerging neurotropic tick-borne flavivirus of increasing public health concern. Overall, this is a highly relevant area of study, and a comprehensive review is well justified.
That said, in my opinion, there are several critical issues that prevent the manuscript from fully achieving its stated goals. The main concern is the lack of a coherent narrative across the text. The manuscript includes many vague or disconnected statements, which obscure the authors’ focus. Below, I include some specific comments to support my overall assessment.
Major Points
Section 1: TLR response to POWV infection and mechanisms of immune evasion
- The authors describe Toll-like receptor (TLR) responses in other viruses and flaviviruses but provide almost no specific information about POWV. Furthermore, Figure 1, titled “a simplified schematic presentation of TLR response to flavivirus infection,” includes receptors that detect DNA intermediates, which are not produced during flavivirus infections.
- Although the stated goal of the manuscript is to describe the innate immune response, the authors dedicate a paragraph to Th1 and Th2 responses in mouse models, which are part of the adaptive immune system and thus fall outside the scope of this review.
- In addition, while the title refers to the innate immune response, the authors focus almost exclusively on TLRs and barely mention RIG-I, which is a key component of the innate antiviral response.
- The acronyms used in Figure 1 are not defined, and the figure itself is not discussed in the main text.
Section 2: POWV vaccines in preclinical and clinical development
- In this section, the authors describe various vaccine platforms targeting POWV. However, this deviates from the main theme of the manuscript. It becomes unclear whether the primary objective is to review the innate immune response to POWV or to discuss vaccine development. Additionally, the potential link between TLR responses and vaccine adjuvants is not clearly addressed—could any of these vaccines be formulated with TLR agonists as adjuvants?
Table 1: TLR agonists in vaccine development
- The purpose of this table is unclear. It is not evident what specific message the authors intend to convey.
Discussion
- Much of the discussion section does not directly relate to the main topic of the review. For example, the authors mention compounds such as NITD008 as potential antivirals. While this is interesting, it contributes little to the central theme of POWV-induced innate immune responses.
- A more critical and integrative discussion is lacking.
- The authors mention that TLR responses during viral infections can act as a double-edged sword, emphasizing the importance of a balanced immune response. This is an important concept. However, its implications—for example, how this duality might influence the use of TLR agonists as vaccine adjuvants—are not explored.
General suggestion: The manuscript would benefit from a clearer definition of its scope and objectives. If the goal is to review the innate immune response to POWV, this should be reflected more consistently throughout the text, with a stronger focus on POWV-specific findings and key innate immune pathways beyond TLRs. Alternatively, if the aim is broader, the sections should be more clearly connected and the narrative better integrated.
Author Response
Major Points
Section 1: TLR response to POWV infection and mechanisms of immune evasion
- The authors describe Toll-like receptor (TLR) responses in other viruses and flaviviruses but provide almost no specific information about POWV. Furthermore, Figure 1, titled “a simplified schematic presentation of TLR response to flavivirus infection,” includes receptors that detect DNA intermediates, which are not produced during flavivirus infections.
Response: We sincerely thank the reviewer for pointing this out and apologize for the oversight. In accordance with the reviewer's suggestion, we have updated Figure 1 accordingly.
Although the stated goal of the manuscript is to describe the innate immune response, the authors dedicate a paragraph to Th1 and Th2 responses in mouse models, which are part of the adaptive immune system and thus fall outside the scope of this review.
Response: We thank the reviewer for this insightful comment. In light of the reviewer’s observation and to maintain the focus of the review, we have removed the paragraph discussing Th1 and Th2 responses to better align the content with the stated scope of the manuscript.
- In addition, while the title refers to the innate immune response, the authors focus almost exclusively on TLRs and barely mention RIG-I, which is a key component of the innate antiviral response.
Response: Thank you for your valuable feedback. We appreciate your observation and in light of your suggestion, we have revised the manuscript including the presumed role of RIG-I (line 251-264).
The acronyms used in Figure 1 are not defined, and the figure itself is not discussed in the main text.
Response: Thank you for pointing this out. We are sorry for the oversight. We have now defined all acronyms used in Figure 1 in the figure legend for clarity. Additionally, we have revised the main text to include a detailed discussion of Figure 1.
Section 2: POWV vaccines in preclinical and clinical development
- In this section, the authors describe various vaccine platforms targeting POWV. However, this deviates from the main theme of the manuscript. It becomes unclear whether the primary objective is to review the innate immune response to POWV or to discuss vaccine development. Additionally, the potential link between TLR responses and vaccine adjuvants is not clearly addressed—could any of these vaccines be formulated with TLR agonists as adjuvants?
Response: We thank the reviewer for this insightful comment. In line with reviewer comments, we have extensively modified the text to fit the context in the manuscript, and to show the potential link between TLR responses and vaccine adjuvants.
Table 1: TLR agonists in vaccine development
- The purpose of this table is unclear. It is not evident what specific message the authors intend to convey.
Response: We thank reviewer for the comment. In response to reviewer comment, we have updated the manuscript and modified as well as moved the table to make relevant and fruitful.
Discussion
- Much of the discussion section does not directly relate to the main topic of the review. For example, the authors mention compounds such as NITD008 as potential antivirals. While this is interesting, it contributes little to the central theme of POWV-induced innate immune responses.
- A more critical and integrative discussion is lacking.
- The authors mention that TLR responses during viral infections can act as a double-edged sword, emphasizing the importance of a balanced immune response. This is an important concept. However, its implications—for example, how this duality might influence the use of TLR agonists as vaccine adjuvants—are not explored.
Response: We have extensively edited the discussion section in response to reviewer all three comments on discussion.
General suggestion: The manuscript would benefit from a clearer definition of its scope and objectives. If the goal is to review the innate immune response to POWV, this should be reflected more consistently throughout the text, with a stronger focus on POWV-specific findings and key innate immune pathways beyond TLRs. Alternatively, if the aim is broader, the sections should be more clearly connected and the narrative better integrated.
Response: We thank the reviewer for highlighting the need for a clearer and more cohesive narrative. In response, we have revised the whole manuscript to satisfy the reviewer comments.

Reviewer 2 Report
Comments and Suggestions for Authors
I overtook the task reviewing this manuscript, because I was interested in innate immune response to Flaviviruses. 80% of TBEV infected persons do not realize their infections, 20% have symptoms, 1-2% have serious consequences or die. Probably the antiviral individual immune response stands behind this phenomenon, most probably because of the effectivity of the innate immune system (IIS). So role of IIS is critical in Flavivirus pathogenesis. Unfortunately I could not pick up any novel information about this topic from this manuscript. The authors write short sentences with a references in the end. A review article is not short sentences with references. If their is any clear result about IIS in Powassan infection, than show it in text, figures tables, and refer. If there are no novel achievements, or any certain information, results about IIS and the Powassan virus, than show this, or do not write review articles. I have seen a general picture about immune response and Flaviviruses, with no details about IIS and the Powassan virus. So the topic is interesting and important, but the manuscript should be reorganized, rewritten, to emphasize the message what I could not found.
Some remarks:
Lines were not numbered
No abbreviations in the Abstract chapter
Discussion
3rd line – geo-graphical
3rd paragraph – in vivo in Italics
Author Response
I overtook the task reviewing this manuscript, because I was interested in innate immune response to Flaviviruses. 80% of TBEV infected persons do not realize their infections, 20% have symptoms, 1-2% have serious consequences or die. Probably the antiviral individual immune response stands behind this phenomenon, most probably because of the effectivity of the innate immune system (IIS). So role of IIS is critical in Flavivirus pathogenesis. Unfortunately I could not pick up any novel information about this topic from this manuscript. The authors write short sentences with a references in the end. A review article is not short sentences with references. If their is any clear result about IIS in Powassan infection, than show it in text, figures tables, and refer. If there are no novel achievements, or any certain information, results about IIS and the Powassan virus, than show this, or do not write review articles. I have seen a general picture about immune response and Flaviviruses, with no details about IIS and the Powassan virus. So the topic is interesting and important, but the manuscript should be reorganized, rewritten, to emphasize the message what I could not found.
Response: We are very grateful to the reviewer for careful reading of the manuscript and comments. In response to reviewer comments, we have reorganized, rewritten and focused on the core message of the review (lines 15-26, 107-132, 136-142, 154-236, 238-242, 253-275, 342-347, 351-385, 589-591, 603-623, 803-821, 825-844, 1218-1245, 1248-1255).
Some remarks:
Lines were not numbered
Response: We have added the line numbers in the revised manuscript.
No abbreviations in the Abstract chapter
Response: Thank you. We have removed any abbreviation used in the abstract.
Discussion
3rd line – geo-graphical
3rd paragraph – in vivo in Italics
Response: Thank you very much. We have extensively edited the discussion section in reply to the comments form another reviewer.

Round 2
Reviewer 1 Report
Comments and Suggestions for Authors
The authors have adequately addressed the issues raised in my previous review, and the manuscript has improved. Still, they should revise the text, mainly regarding the use of abbreviations.
– Terms should be defined only at their first mention. Some examples:
Line 215: use TLRs rather than Toll-like receptors
Line 305: use PWV rather than Powassan virus
Line 450: use PAMPs rather than pathogen-associated molecular patterns
Line 735: use VLP rather than virus-like particle
Author Response
Comments and Suggestions for Authors
The authors have adequately addressed the issues raised in my previous review, and the manuscript has improved. Still, they should revise the text, mainly regarding the use of abbreviations.
– Terms should be defined only at their first mention. Some examples:
Line 215: use TLRs rather than Toll-like receptors
Line 305: use PWV rather than Powassan virus
Line 450: use PAMPs rather than pathogen-associated molecular patterns
Line 735: use VLP rather than virus-like particle
Response: We sincerely thank the reviewer for the positive evaluation of our revised manuscript and for the suggestions regarding the use of abbreviations. We have carefully revised the text to ensure that all terms are defined only at their first mention and have implemented the suggested changes at the indicated lines (e.g., TLRs, PWV, PAMPs, and VLP).
Reviewer 2 Report
Comments and Suggestions for Authors
As the authors reorganised the text in many points, improving the manuscript, I suggest its publication in its present form.
Author Response
Comments and Suggestions for Authors
As the authors reorganised the text in many points, improving the manuscript, I suggest its publication in its present form.
Response: We sincerely thank the reviewer for the positive evaluation and recommendation for publication of the manuscript in its current form.